# Mutant mice lacking alternatively spliced p53 isoforms unveil *Ackr4* as a male-specific prognostic factor in Myc-driven B-cell lymphomas

Anne Fajac[1,2,3,4], Iva Simeonova[1,2,3,4], Julia Leemput[1,2,3,4], Marc Gabriel[2,3,4,5], Aurélie Morin[1,2,3,4], Vincent Lejour[1,2,3,4], Annaïg Hamon[1,2,3,4], Jeanne Rakotopare[1,2,3,4], Wilhelm Vaysse-Zinkhöfer[1,2,3,4], Eliana Eldawra[1,2,3,4], Marina Pinskaya[2,3,4,5], Antonin Morillon[2,3,4,5], Jean-Christophe Bourdon[6], Boris Bardot[1,2,3,4]*, Franck Toledo[1,2,3,4]*

[1]Genetics of Tumor Suppression, Institut Curie, Paris, France; [2]CNRS UMR3244, Paris, France; [3]Sorbonne University, Paris, France; [4]PSL Research University, Paris, France; [5]Non Coding RNA, Epigenetic and Genome Fluidity, Institut Curie, Paris, France; [6]School of Medicine, Ninewells Hospital, University of Dundee, Dundee, United Kingdom

**\*For correspondence:**
boris.bardot@curie.fr (BB);
Franck.Toledo@curie.fr (FT)

**Competing interest:** The authors declare that no competing interests exist.

**Abstract** The *Trp53* gene encodes several isoforms of elusive biological significance. Here, we show that mice lacking the *Trp53* alternatively spliced (AS) exon, thereby expressing the canonical p53 protein but not isoforms with the AS C-terminus, have unexpectedly lost a male-specific protection against Myc-induced B-cell lymphomas. Lymphomagenesis was delayed in *Trp53*^(+/+)*Eμ-Myc* males compared to *Trp53*^(ΔAS/ΔAS) *Eμ-Myc* males, but also compared to *Trp53*^(+/+)*Eμ-Myc* and *Trp53*^(ΔAS/ΔAS) *Eμ-Myc* females. Pre-tumoral splenic cells from *Trp53*^(+/+)*Eμ-Myc* males exhibited a higher expression of *Ackr4*, encoding an atypical chemokine receptor with tumor suppressive effects. We identified *Ackr4* as a p53 target gene whose p53-mediated transactivation is inhibited by estrogens, and as a male-specific factor of good prognosis relevant for murine *Eμ-Myc*-induced and human Burkitt lymphomas. Furthermore, the knockout of *ACKR4* increased the chemokine-guided migration of Burkitt lymphoma cells. These data demonstrate the functional relevance of alternatively spliced p53 isoforms and reveal sex disparities in Myc-driven lymphomagenesis.

## eLife assessment

This **important** study using engineered mouse models provides a first and **compelling** demonstration of a pathogenic phenotype associated with lack of expression of p53AS isoforms, isoforms of the p53 protein with a different C-terminus than canonical p53. The role of these isoforms has been elusive so far and this first demonstration represents a substantial advance in our understanding of the complex role(s) of p53 isoforms. The revised article adequately addresses previous concerns.

## Introduction

*TP53*, the human gene for tumor suppressor p53, encodes several isoforms owing to distinct promoters, alternative splicing, and multiple translation initiation sites (*Bourdon et al., 2005*; *Courtois et al., 2002*; *Flaman et al., 1996*; *Yin et al., 2002*). p53 alternative isoforms can be abnormally expressed in cancer cells and some may regulate the canonical p53 protein (*Anbarasan and Bourdon,*

**eLife digest** Human cells divide many times during a lifetime, a process that requires careful regulation to avoid uncontrolled cell division, which can lead to various disorders, including cancer. For example, *TP53*, which encodes multiple proteins, is the most commonly mutated gene in cancers.

*TP53* carries the instructions to make a tumor suppressor protein, known as p53, which can stop cancers from forming and spreading. In humans and mice, *TP53* (and the mouse analogue *Trp53*) can also be read by the cell to make several slightly different versions of the p53 protein, known as isoforms. The p53 isoforms are much less studied and their role in an organism is still unclear.

To address this, Fajac et al. used genome editing to make mouse strains that were still able to express p53, but were only able to create a specific subset of p53 isoforms. In these mice, part of the *Trp53* gene had been mutated to remove the cell's ability to make isoforms with an alternative C-terminal end.

Fajac et al. then allowed these mice to breed with mice that were model organisms for a cancer called B-cell lymphoma. This revealed that male offspring that lacked alternative p53 isoforms were more susceptible to B-cell lymphoma and that they had decreased levels of the protein ACKR4, a receptor for signaling proteins that regulate cellular movement. Human datasets showed that having higher levels of ACKR4 could be linked to a better disease prognosis in male patients with Burkitt lymphoma, a rare but aggressive form of B-cell lymphoma. The same effect was not observed in females, suggesting that measuring ACKR4 gene expression in male patients with Burkitt lymphoma might be useful to identify the patients at higher risk.

The study from Fajac et al. provides a new perspective on p53 – one of the most studied proteins. It highlights specific p53 isoforms and the ACKR4 protein as a potential way to identify male patients at higher risk from a type of B-cell lymphoma.

---

*2019*; *Bourdon et al., 2005*; *Mondal et al., 2013*; *Senturk et al., 2014*). However, aberrant RNA splicing is a common feature of cancer cells (*Graubert et al., 2012*; *Martin et al., 2013*; *Pajares et al., 2007*; *Sette and Paronetto, 2022*) and to which extent alternative splicing generates functionally relevant proteins is controversial (*Abascal et al., 2015*; *Bardot and Toledo, 2015*; *Blencowe, 2017*; *Tress et al., 2017a*; *Tress et al., 2017b*; *Ule and Blencowe, 2019*; *Weatheritt et al., 2016*). Thus, the biological importance of many p53 isoforms remains elusive.

Like its human *TP53* homolog, the murine *Trp53* gene encodes multiple isoforms differing in their N- or C-termini (*Arai et al., 1986*; *Marcel et al., 2011*). Mouse models to evaluate the role of p53 isoforms differing in their N-terminus revealed that Δ40-p53 overexpression leads to accelerated ageing (*Maier et al., 2004*; *Steffens Reinhardt et al., 2020*). However, the potential role of p53 isoforms with an alternative C-terminus was not analyzed in vivo. p53 isoforms with distinct C-termini result from the splicing of two mutually exclusive final exons: exon 11, encoding the canonical 'α' C-terminal domain, and the alternatively spliced (AS) exon, encoding another C-terminus (*Arai et al., 1986*). In adult mice, isoforms with the canonical C-terminus are predominant in all tissues (*Figure 1—figure supplement 1A*). Two models (*Trp53*$^{\Delta 31}$ and *Trp53*$^{\Delta CTD}$), designed to study the consequences of a loss of the canonical p53 C-terminus, exhibited signs of increased p53 activity, leading to a rapidly lethal anemia (*Hamard et al., 2013*; *Simeonova et al., 2013*). To determine the role of p53-AS isoforms in vivo, we created *Trp53*$^{\Delta AS}$, a mouse model with a specific deletion of the AS exon (*Figure 1—figure supplement 1B*). In mouse embryonic fibroblasts (MEFs), the *Trp53*$^{\Delta AS}$ allele prevented the expression of isoforms with the AS C-terminus, whereas it did not affect RNA levels for p53 isoforms with the canonical C-terminus (*Figure 1—figure supplement 1C*). We previously used this model to show that p53-AS isoforms had no role in the anemia affecting *Trp53*$^{\Delta 31/\Delta 31}$ mice (*Simeonova et al., 2013*). However, a detailed phenotyping of *Trp53*$^{\Delta AS/\Delta AS}$ mice remained to be performed. The detailed phenotyping, presented here, yielded surprising information on lymphomagenesis.

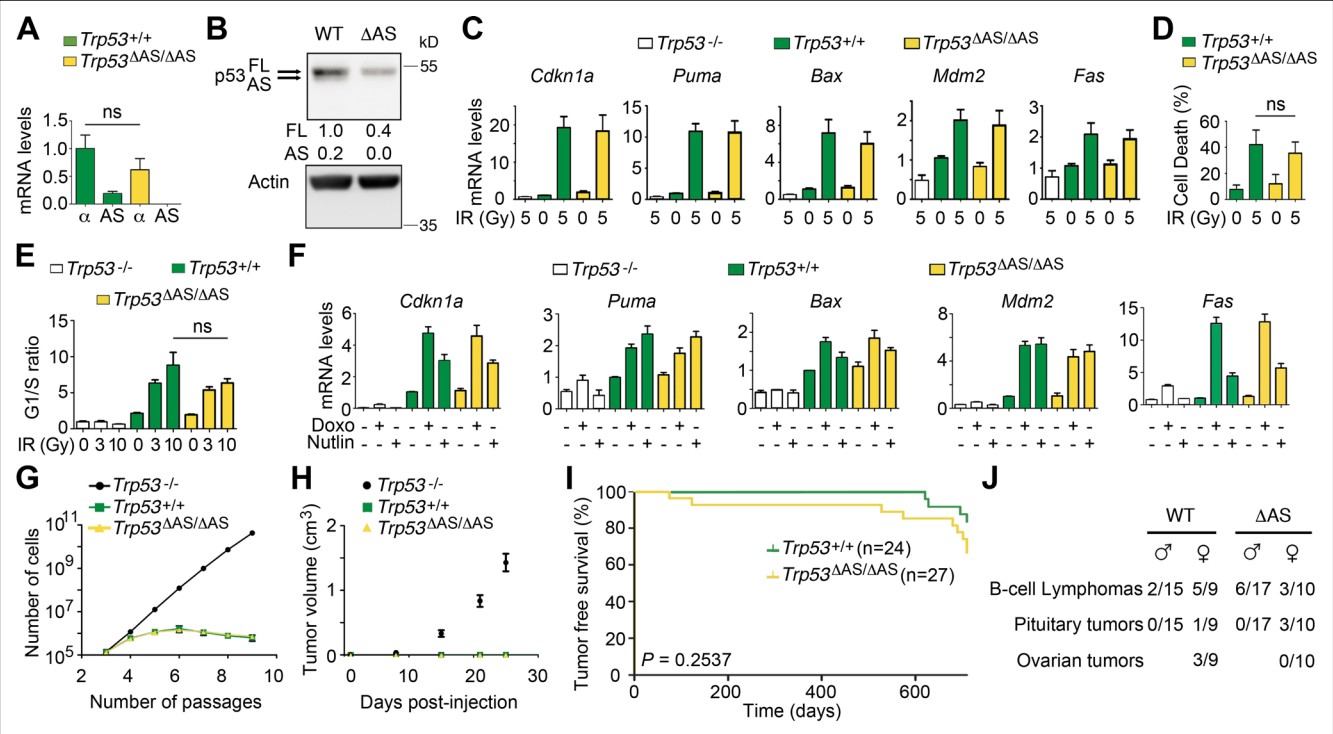

**Figure 1.** The loss of p53-AS isoforms does not alter cellular stress responses or survival to spontaneous tumors. (**A**) mRNAs for p53-α and p53-AS isoforms from thymocytes of irradiated mice were quantified by RT-qPCR, and p53-α levels in *Trp53*$^{+/+}$ mice were assigned a value of 1. Means ± SEM (n = 3). (**B**) Protein extracts from thymocytes of *Trp53*$^{+/+}$ (WT) or *Trp53*$^{ΔAS/ΔAS}$ (ΔAS) irradiated mice were immunoblotted with p53 or actin antibodies. After normalization to actin, full-length (FL) p53-α levels in WT thymocytes were assigned a value of 1. (**C**) mRNA levels of p53 target genes in thymocytes, before or after γ-irradiation. Means ± SEM (n = 3). (**D**) Thymocyte apoptotic response to γ-irradiation. Means ± SEM (n = 6). (**E**) Cell cycle control in mouse embryonic fibroblasts (MEFs) after γ-irradiation. Asynchronous MEFs were exposed to 0–10 Gy γ-irradiation, and after 24 hr, cells were labeled with BrdU for 1 hr and analyzed by FACS. Means ± SEM from >3 independent experiments with at least two independent MEF clones per genotype. (**F**) mRNA levels of p53 target genes in MEFs untreated or treated with 0.5 µg/ml of clastogenic doxorubicin (Doxo) or 10 µM of the Mdm2 antagonist Nutlin. Means ± SEM from >3 experiments with ≥2 independent MEF clones. (**G**) MEFs proliferation under hyperoxic conditions. Cells were grown according to a 3T3 protocol. Each point is the mean from four independent MEF clones, the value for each clone resulting from triplicates. (**H**) Growth of tumor xenografts. E1A+Ras-expressing MEFs were injected into the flanks of nude mice and tumor volumes were determined after 1–25 days. Means ± SD (n = 4 per timepoint and genotype). (**I**) Tumor-free survival of *Trp53*$^{+/+}$ and *Trp53*$^{ΔAS/ΔAS}$ mice (n = cohort size). (**J**) Incidence of the indicated tumor types, determined at death after macroscopic examination and histological analysis of *Trp53*$^{+/+}$ (WT) and *Trp53*$^{ΔAS/ΔAS}$ (ΔAS) mice. In (**A, D, E**), ns = non-significant in Student's *t*-test.

The online version of this article includes the following source data and figure supplement(s) for figure 1:

**Source data 1.** Raw unedited gels and blots for *Figure 1B*.

**Source data 2.** Uncropped and labeled gels and blots for *Figure 1B*.

**Figure supplement 1.** Description of the *Trp53*$^{ΔAS}$ mouse model and analysis of *Trp53*$^{ΔAS/ΔAS}$ thymocytes and fibroblasts.

## Results

### Stress responses in WT and *Trp53*$^{ΔAS/ΔAS}$ cells

We analyzed cellular stress responses in thymocytes, known to undergo a p53-dependent apoptosis upon irradiation (*Lowe et al., 1993*), and in primary fibroblasts, known to undergo a p53-dependent cell cycle arrest in response to various stresses; for example, DNA damage caused by irradiation or doxorubicin (*Kastan et al., 1992*), and the Nutlin-mediated inhibition of Mdm2, a negative regulator of p53 (*Vassilev et al., 2004*). We first compared thymocytes from irradiated wild-type (WT) and *Trp53*$^{ΔAS/ΔAS}$ mice. In WT thymocytes, isoforms with the AS C-terminus were five times less abundant than isoforms with the α C-terminus at the RNA level (*Figure 1A*), and in western blots the p53-AS protein appeared as a faint band running just ahead of, and often hard to separate from, the band specific for p53-α, the canonical full-length p53 (*Figure 1B*). In *Trp53*$^{ΔAS/ΔAS}$ thymocytes, mRNA levels for α isoforms were slightly decreased, if at all (*Figure 1A*), whereas p53-α protein levels appeared

markedly decreased (*Figure 1B*), raising the possibility that p53-AS isoforms might contribute to p53-α abundance. Nevertheless, the transactivation of classical p53 target genes (*Figure 1C*) and apoptotic response (*Figure 1D*, *Figure 1—figure supplement 1D*) were not significantly altered by the loss of AS isoforms. Likewise, no significant difference was observed between WT and *Trp53^{ΔAS/ΔAS}* fibroblasts in assays for cell cycle control (*Figure 1E*, *Figure 1—figure supplement 1E*), expression of well-known p53 target genes (*Figure 1F*, *Figure 1—figure supplement 1F and G*), proliferation under hyperoxic conditions (*Figure 1G*), or the growth of tumor xenografts (*Figure 1H*).

## Lymphomagenesis in WT and *Trp53^{ΔAS/ΔAS}* mice

We compared spontaneous tumor onset in WT and *Trp53^{ΔAS/ΔAS}* littermates for over 2 years and observed no significant difference in tumor-free survival (*Figure 1I*). Because lymphoma is a common neoplasm in C57Bl/6J WT mice (*Brayton et al., 2012*) and our mouse cohorts resulted from >10 generations of backcrosses with C57Bl/6J mice, we searched for evidence of lymphoma in the lymph nodes and spleen, by macroscopic examination at autopsy and histological analyses. B-cell lymphomas were observed in about 30% of mice of either genotype (*Figure 1J*). In WT mice, a higher incidence of

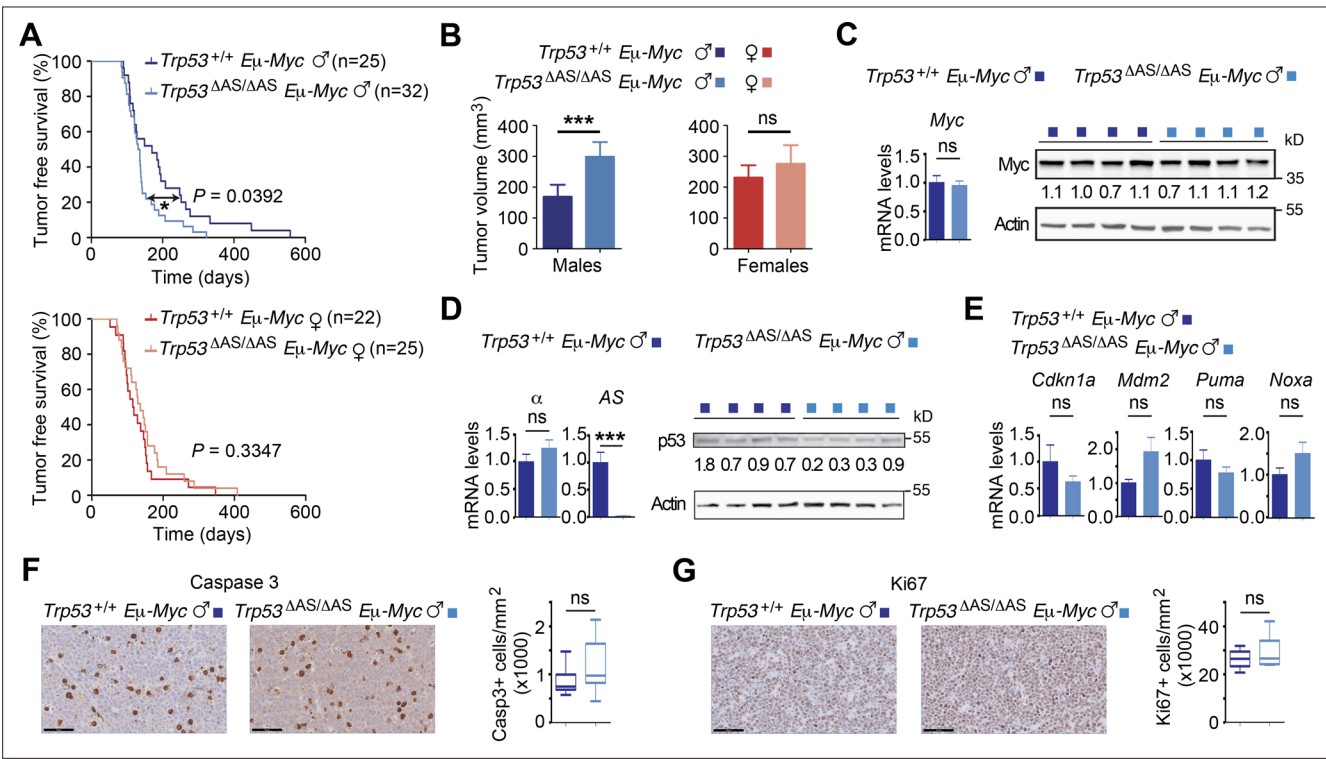

**Figure 2.** Male-specific acceleration of Myc-induced B-cell lymphomagenesis in mice lacking p53-AS isoforms. (**A**) Tumor-free survival of *Trp53^{+/+} Eμ-Myc* and *Trp53^{ΔAS/ΔAS} Eμ-Myc* mice, classified according to sex (n = cohort size). (**B**) Tumor volumes upon dissection of *Trp53^{+/+} Eμ-Myc* and *Trp53^{ΔAS/ΔAS} Eμ-Myc* mice, classified according to sex. Means ± SEM from 100 lymph nodes from *Trp53^{+/+} Eμ-Myc* males, 96 from *Trp53^{+/+} Eμ-Myc* females, 148 from *Trp53^{ΔAS/ΔAS} Eμ-Myc* males, and 124 from *Trp53^{ΔAS/ΔAS} Eμ-Myc* females. (**C**) Myc mRNA and protein levels in lymph node tumors. (**D**) Levels of p53-α and p53-AS transcripts and p53 protein levels in lymph node tumors. (**E**) Transcript levels of the indicated p53 target genes. Means ± SEM (n = 6 per genotype). (**F, G**) Apoptosis (**F**) and cell proliferation (**G**) in tumor lymph nodes from *Eμ-Myc* males were determined by immunohistochemistry with antibodies against cleaved caspase-3 and ki67, respectively. Positive cells were counted and normalized to the analyzed areas. Means ± SEM (n = 6 mice per assay and genotype). In (**F, G**) scale bars = 50 μm. Statistical analyses with Mantel–Cox (**A**) and Student's *t* (**B–G**) tests. ***p<0.001, *p<0.05, ns: nonsignificant.

The online version of this article includes the following source data and figure supplement(s) for figure 2:

**Source data 1.** Raw unedited gels and blots for *Figure 2C*.

**Source data 2.** Raw unedited gels and blots for *Figure 2D*.

**Source data 3.** Uncropped and labeled gels and blots for *Figure 2C*.

**Source data 4.** Uncropped and labeled gels and blots for *Figure 2D*.

**Figure supplement 1.** Analysis of tumors from *Trp53^{+/+} Eμ-myc* and *Trp53^{ΔAS/ΔAS} Eμ-Myc* mice.

B-cell lymphomas was observed in females, in agreement with previous observations (***Brayton et al., 2012***). By contrast, no obvious sex-specific bias was observed for B-cell lymphomas in *Trp53^{ΔAS/ΔAS}* mice (***Figure 1J***), raising the possibility that the loss of p53-AS isoforms affected B-cell lymphomagenesis. However, the numbers of lymphoma-bearing mice were too small to be conclusive.

We next used *Eμ-Myc* transgenic mice, prone to highly penetrant B-cell lymphomas (***Adams et al., 1985***). *Trp53^{+/+} Eμ-Myc* and *Trp53^{ΔAS/ΔAS} Eμ-Myc* mice developed B-cell lymphomas (***Figure 2—figure supplement 1A***) with similar survival curves when sexes were not considered (***Figure 2—figure supplement 1B***). Importantly, however, death was accelerated, and tumor lymph nodes were larger, in *Trp53^{ΔAS/ΔAS} Eμ-Myc* males compared to their *Trp53^{+/+} Eμ-Myc* male counterparts, whereas no difference in lymphomagenesis was noticeable between *Trp53^{ΔAS/ΔAS} Eμ-Myc* and *Trp53^{+/+} Eμ-Myc* female mice (***Figure 2A and B***). Our data (***Figure 2A and B***, ***Figure 2—figure supplement 1C***), together with the fact that B-cell lymphomas occur with a higher incidence in WT C57Bl/6J female mice (***Brayton et al., 2012***), suggested that *Trp53^{+/+} Eμ-Myc* male mice are more refractory to B-cell lymphomas, and that p53-AS isoforms might confer this male-specific protection against lymphomagenesis.

## Cause for accelerated lymphomagenesis in *Trp53^{ΔAS/ΔAS} Eμ-Myc* males

We next aimed to determine the mechanisms underlying the accelerated lymphomagenesis in *Trp53^{ΔAS/ΔAS} Eμ-Myc* males. Inactivating p53 mutations were not more frequent in tumors from *Trp53^{ΔAS/ΔAS} Eμ-Myc* males than in those from *Trp53^{+/+} Eμ-Myc* males, ruling out additional mutations at the *Trp53* locus as potential causes for accelerated lymphomagenesis in *Trp53^{ΔAS/ΔAS} Eμ-Myc* males (***Figure 2—figure supplement 1D and E***). We next analyzed a subset of tumors with no detectable *Trp53* mutation in males of both genotypes and found that Myc was expressed at similar RNA and protein levels in all tumors (***Figure 2C***). No difference in p53-α mRNA levels was observed in tumors from both genotypes, although a decrease at the protein level was detected in most tumors from *Trp53^{ΔAS/ΔAS} Eμ-Myc* males (***Figure 2D***). Nevertheless, similar transcript levels for classical p53 target genes were observed in tumor cells of both genotypes (***Figure 2E***). To test whether a higher tumor volume in *Trp53^{ΔAS/ΔAS} Eμ-myc* males might result from lower apoptosis and/or higher cell proliferation, we next analyzed tumors by immunohistochemistry with antibodies against cleaved caspase-3 or ki-67, respectively. Similar apoptotic and proliferation indexes were observed for both genotypes (***Figure 2F and G***). In sum, classical assays for p53 activity in tumors failed to account for differences between the two male genotypes.

The speed of lymphomagenesis in *Eμ-Myc* mice correlates with the extent of B-cell expansion in the first stages of B-cell differentiation (***Langdon et al., 1986***) and p53 was proposed to control the pool of early B cells (***Slatter et al., 2010***). Therefore, we determined the levels of the early pre-B/immature B cells in 6-week-old mice, before any sign of tumor. We analyzed the spleen, a preferential site of B-cell expansion (***Langdon et al., 1986***) with a relatively high AS/α isoform ratio (***Figure 1—figure supplement 1A***). Flow cytometry with a combination of markers was used to discriminate the pre-B, immature, transitional, and mature B subpopulations. As expected (***Langdon et al., 1986***), we observed high numbers of pre-B and immature B cells in *Eμ-Myc* mice. In males, pre-B and immature B cells were more abundant in *Trp53^{ΔAS/ΔAS} Eμ-Myc* animals, while no difference was observed for transitional and mature B cells (***Figure 3A***, ***Figure 3—figure supplement 1A***). By contrast, in the spleen of *Trp53^{+/+}* and *Trp53^{ΔAS/ΔAS}* 6-week-old male mice without the *Eμ-Myc* transgene, most B cells were mature B cells (***Figure 3—figure supplement 1B***). In *Eμ-Myc* females, the numbers of pre-B and immature B cells were similar between genotypes, as were the numbers of transitional and mature B cells (***Figure 3A***). Interestingly, *Trp53^{+/+} Eμ-myc* males, which develop lymphomas less rapidly, exhibited the lowest number of immature B cells (***Figure 3A***), suggesting a direct correlation between the level of immature B cell expansion and the speed of lymphomagenesis. Together, these data suggested that p53-AS isoforms may not be required to control the pool of early B cells under normal conditions, but that in an *Eμ-Myc* context they would limit the expansion of pre-tumor early B cells, specifically in males.

## Transcriptomes from *Trp53^{+/+} Eμ-Myc* and *Trp53^{ΔAS/ΔAS} Eμ-Myc* male spleens

We next performed bulk RNA-seq and differential expression analyses comparing the spleens from 4- to 6-week-old *Trp53^{ΔAS/ΔAS} Eμ-Myc* males to spleens from age-matched *Trp53^{+/+} Eμ-Myc* males. This

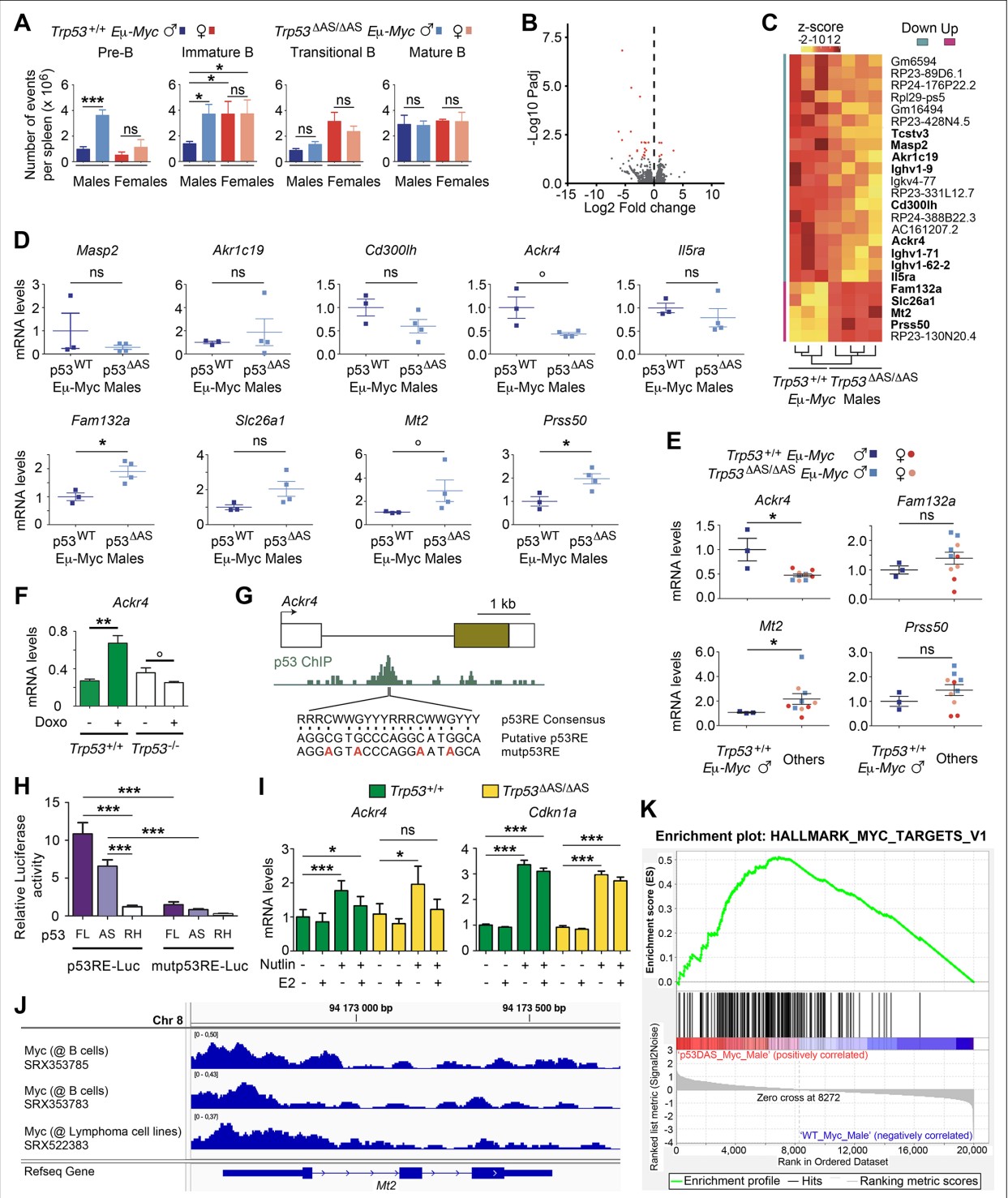

**Figure 3.** The loss of p53-AS isoforms affects *Ackr4* expression in *Eμ-Myc* male mice. (**A**) B-cell subpopulations in spleens of 6-week-old *Trp53⁺/⁺ Eμ-Myc* and *Trp53^ΔAS/ΔAS Eμ-Myc* mice. Means ± SEM (n = 6 per genotype). (**B, C**) RNAseq analysis of spleens from *Trp53⁺/⁺ Eμ-Myc* (n = 3) and *Trp53^ΔAS/ΔAS Eμ-Myc* (n = 4) 4–6-week-old male mice. Volcano plot (**B**), with differentially expressed genes (DEGs) in red. Unsupervised clustering heatmap plot (**C**), with DEGs ranked according to mean fold changes, and protein-coding genes in bold. (**D**) RT-qPCR analysis of candidate DEGs from spleens of *Trp53⁺/⁺* (p53^WT) *Eμ-Myc* males and *Trp53^ΔAS/ΔAS* (p53^ΔAS) *Eμ-Myc* males. Means ± SEM (n = 3–4 per genotype). (**E**) RT-qPCR analysis of indicated DEGs from spleens of 4–6-week-old *Trp53⁺/⁺ Eμ-Myc* males, *Trp53^ΔAS/ΔAS Eμ-Myc* males, *Trp53⁺/⁺ Eμ-Myc* females, and *Trp53^ΔAS/ΔAS Eμ-Myc* females. Means ± SEM (n = 3–4 per sex and genotype). (**F**) *Ackr4* is transactivated by p53 in response to stress. Ackr4 mRNAs in untreated or doxorubicin-treated WT and *Trp53⁻/⁻* mouse embryonic fibroblasts (MEFs). Data from 2 to 3 MEFs per genotype (**Younger et al., 2015**). (**G**) A putative p53 response element in *Ackr4*

*Figure 3 continued on next page*

*Figure 3 continued*

intron 1. Top: map of the *Ackr4* gene. (boxes: exons [brown box: translated region]; black line: intron 1); middle: p53 ChIP in doxorubicin-treated MEFs according to ChIP-Atlas (SRX270554) (*Oki et al., 2018*); bottom: p53 Response Element (p53RE) consensus sequence (R = G or A, W = A or T, Y = C or T), the putative p53RE and its mutated counterpart. (**H**) Luciferase assays of the candidate p53RE. A 1.5 kb fragment containing the WT or mutant p53RE was cloned upstream a luciferase reporter, then transfected into *Trp53⁻/⁻* MEFs together with an expression plasmid for full length p53 (FL), p53-AS or the DNA-binding mutant p53^R270H (RH). Means ± SEM (n = 4–6). (**I**) In MEFs, p53 activation leads to an increased *Ackr4* expression attenuated by estradiol. *Ackr4* and *Cdkn1a* mRNAs were quantified by RT-qPCR from *Trp53⁺/⁺* and *Trp53^ΔAS/ΔAS* MEFs, untreated or treated with 10 μM Nutlin and/or 5 μg/ml 17-β estradiol (E2). Means ± SEM from four independent experiments. (**J**) Evidence for Myc binding at the *Mt2* promoter in B cells. ChIP-Atlas reports Myc binding to *Mt2* promoter sequences in primary B cells from the lymph nodes of Eμ-Myc mice (SRX353785, SRX353783) and in Eμ-Myc-induced lymphoma cells (SRX522383). Chr: chromosome. (**K**) Gene set enrichment analysis (GSEA). GSEA, performed in *Trp53⁺/⁺* Eμ-Myc (WT_Myc) and *Trp53^ΔAS/ΔAS* Eμ-Myc (p53DAS_Myc) male splenic cells, indicated an enrichment of hallmark Myc targets in *Trp53^ΔAS/ΔAS* Eμ-Myc cells. In (**A, D, E, F, H, I**) **p<0.001, **p<0.01, *p<0.05, °p≤0.057, ns: nonsignificant in Student's *t* or Mann–Whitney tests.

The online version of this article includes the following source data and figure supplement(s) for figure 3:

**Figure supplement 1.** Analysis of pre-tumoral spleens.

**Figure supplement 1—source data 1.** Raw unedited gels and blots for *Figure 3—figure supplement 1C*.

**Figure supplement 1—source data 2.** Uncropped and labeled gels and blots for *Figure 3—figure supplement 1C*.

revealed a limited number of significantly deregulated genes (*Figure 3B*), including 13 protein-coding genes and 11 pseudogenes (*Figure 3C*). Out of the 13 protein-coding genes, we focused on the 10 genes not encoding an immunoglobulin and analyzed the same samples by RT-qPCR (*Figure 3D*). For 6 of the 10 genes, expression levels were too low to be quantified (*Tcstv3*), or differences in expression were not statistically significant (*Masp2, Akr1c19, Cd300lh, Il5ra, Slc26a1*). Of note, *Il5ra* belonged to this group, although it is regulated by p53 (*Zhu et al., 2022*), which illustrates the difficulty to analyze subtle effects in our experiments. Taking this into account, we considered as potentially interesting the four remaining genes, exhibiting differences in mRNA levels with statistical significance (p<0.05) or borderline statistical significance (p=0.057): *Ackr4*, less expressed in *Trp53^ΔAS/ΔAS* Eμ-Myc males, and *Fam132a, Mt2,* and *Prss50*, with an increased expression in *Trp53^ΔAS/ΔAS* Eμ-Myc males (*Figure 3D*). Importantly, survival curves indicated that the Myc-induced lethality was delayed in *Trp53⁺/⁺* Eμ-Myc males compared to *Trp53^ΔAS/ΔAS* Eμ-Myc males, *Trp53⁺/⁺* Eμ-Myc females, and *Trp53^ΔAS/ΔAS* Eμ-Myc females (*Figure 2—figure supplement 1C*). Thus, we quantified transcripts for *Ackr4, Fam132a, Mt2,* and *Prss50* in the spleen of 4–6-week-old *Trp53⁺/⁺* Eμ-Myc and *Trp53^ΔAS/ΔAS* Eμ-Myc females, then compared mRNA levels in *Trp53⁺/⁺* Eμ-Myc males versus the three other groups. Significantly higher expression of *Ackr4* and lower expression of *Mt2* were found in *Trp53⁺/⁺* Eμ-Myc males (*Figure 3E*).

*Ackr4* (also known as *Ccrl1*) encodes the atypical chemokine receptor 4, a decoy receptor promoting the degradation of chemokines that modulate cancer cell proliferation and metastasis (*Chow and Luster, 2014*; *Marcuzzi et al., 2018*; *Müller et al., 2001*). Our data suggested that *Ackr4* might be a gene transactivated by p53-α and/or p53-AS isoforms. Consistent with this, by extracting data from a transcriptome-wide study in MEFs (*Younger et al., 2015*), we found evidence for a p53-dependent transactivation of *Ackr4* in response to doxorubicin (*Figure 3F*). Furthermore, ChIP-Atlas, the database of chromatin immunoprecipitation experiments (*Oki et al., 2018*), indicated p53 binding to sequences within the intron 1 of *Ackr4* in doxorubicin-treated MEFs, and we identified a candidate p53 responsive element in this intron (*Figure 3G*). We next used luciferase assays to show that this p53 responsive element can be bound and regulated by both p53-α and p53-AS (*Figure 3G and H*, *Figure 3—figure supplement 1C*). Together, these data show that *Ackr4* is indeed a p53 target gene, although RNAseq data indicated that it is expressed at much lower levels than classical p53 targets like *Cdkn1a* or *Mdm2* in the splenic cells of Eμ-Myc mice (*Supplementary file 1*). Furthermore, *Ackr4* was shown to be regulated by Foxl2 and estrogen signaling in ovarian cells (*Georges et al., 2014*) and 17-β estradiol was recently found to regulate *ACKR4* expression in meniscal cells from both sexes, albeit differentially (*Knewtson et al., 2020*). Accordingly, we observed, in both WT and *Trp53^ΔAS/ΔAS* MEFs, that p53 activation with the Mdm2 antagonist Nutlin led to the transactivation of *Ackr4*, but that a concomitant treatment with 17-β estradiol markedly decreased, or completely abrogated, *Ackr4* transactivation (*Figure 3I*). By contrast, *Cdkn1a* was efficiently transactivated under both conditions in mutant and WT cells (*Figure 3I*). These data indicate that *Ackr4* is a p53 target gene whose p53-mediated transactivation can be inhibited by estrogens.

The *Mt2* gene, encoding the potentially oncogenic metallothionein-2 (*Si and Lang, 2018*), was less expressed in *Trp53*[+/+] *Eμ-Myc* male pre-tumoral splenic cells, which raised the possibility of its direct or indirect repression by p53, potentially through the binding of p53 or the DREAM complex at its promoter (*Engeland, 2018*; *Peuget and Selivanova, 2021*). However, ChIP-Atlas reported no binding of these proteins at the *Mt2* promoter. Alternatively, evidence that Myc may impact on *Mt2* expression was obtained previously (*Qin et al., 2021*), and ChIP-Atlas reported Myc binding at the *Mt2* promoter in primary B cells from lymph nodes of *Eμ-Myc* mice as well as *Eμ-Myc*-induced lymphoma cells (*Figure 3J*). This may suggest that the lower expression of *Mt2* in pre-tumoral splenic cells from *Trp53*[+/+] *Eμ-Myc* males (*Figure 3E*) might result from a subtle difference in Myc signaling. Consistent with this, the comparison of transcriptomes of splenic cells from *Trp53*[ΔAS/ΔAS] *Eμ-Myc* males and *Trp53*[+/+] *Eμ-Myc* males, when analyzed by gene set enrichment analysis (*Subramanian et al., 2005*), revealed an enrichment of hallmark Myc target genes in *Trp53*[ΔAS/ΔAS] *Eμ-Myc* male splenic cells (*Figure 3K*, *Supplementary file 2*).

## Relevance of *ACKR4* expression in Burkitt lymphomas

Murine and human alternatively spliced p53 isoforms exhibit structural differences (*Marcel et al., 2011*) and the ChIP-Atlas database (*Oki et al., 2018*) does not report p53 binding to the human *ACKR4* intron 1. Nevertheless, we found that p53 activation in human cells also led to an increased *ACKR4* expression abrogated by 17-β estradiol, whereas 17-β estradiol had no significant effect on the p53-mediated transactivation of *CDKN1A* (*Figure 4A*). This led us to investigate the potential relevance of *ACKR4* expression in human B-cell lymphomas. We analyzed public databases of B-cell lymphomas patients with clinical and gene expression information. We first analyzed #GSE4475 (*Hummel et al., 2006*), a dataset of mature aggressive B-cell lymphomas previously used to define Burkitt lymphoma-like specific transcriptomes, comprising 159 patients (91 men, 68 women) with clinical follow-up. Overall, *ACKR4* gene expression was not significantly different between male and female patients (*Figure 4B*, left). However, average mRNA levels appeared higher in males when we considered the 30% patients of each sex with the highest *ACKR4* expression (*Figure 4B*, right). Strikingly, when we compared the survival of 30% patients with the highest ACKR4 mRNA levels to the survival of the 30% patients with the lowest ACKR4 mRNA levels, high *ACKR4* expression correlated with a better prognosis in men, but not in women (*Figure 4C*). By contrast, for *MT2A*, the human homolog of *Mt2*, differences in mRNA levels did not correlate with significant differences in survival for either sex (*Figure 4—figure supplement 1A*).

We also analyzed dataset #GSE181063 (*Lacy et al., 2020*), comprising mostly diffuse large B-cell lymphomas (DLBCL; 613 men, 536 women) and a few Burkitt lymphomas (65 men, 18 women). We found no difference in survival curves of DLBCL patients with low versus high *ACKR4* levels, neither in men nor in women. However, there was again an increased survival for Burkitt lymphoma male patients with high *ACKR4* expression, but not for women (*Figure 4—figure supplement 1B*). Next, we analyzed #phs000235 (*Morin et al., 2011*), a Burkitt lymphoma-specific dataset (65 men, 37 women) comprising mostly patients diagnosed at 0–17 years of age, hence providing cohorts homogeneous for both tumor type and age of onset. Again, *ACKR4* was expressed at higher levels in a subset of male patients (*Figure 4D*) and high *ACKR4* expression correlated with a better prognosis only in males (*Figure 4E*). Finally, we analyzed dataset #GSE136337 (*Danziger et al., 2020*), comprising data from the malignant plasma cells of patients with multiple myeloma (260 men, 166 women), and found that *ACKR4* is not a prognostic factor for this cancer type (*Figure 4—figure supplement 1C*). Altogether, these analyses led us to conclude that, as in *Eμ-Myc* mice, *ACKR4* is a male-specific positive prognostic factor in Burkitt lymphoma, the archetype of MYC-driven B-cell lymphomas.

We next considered the possibility that ACKR4 might be more than a biomarker, if it acts as a suppressor of MYC-driven B-cell lymphomagenesis. In support of this hypothesis, ACKR4 scavenges the chemokine CCL21, a ligand of the chemokine receptor CCR7 (*Bastow et al., 2021*; *Ulvmar et al., 2014*), and the ACKR4-mediated sequestration of CCL21 may impair the CCR7 signaling cascade, which might lead to decreased MYC activity (*Shi et al., 2015*). Consistent with this, a *Ccr7* deficiency was shown to delay the lymphomagenesis induced by *Eμ-Myc* in mice (*Rehm et al., 2011*). In addition, Ccr7 is required for lymphoma cell lodging to secondary lymphoid organs (*Rehm et al., 2011*) and ACKR4 expression was inversely correlated with the metastasis capacity of different types of cancer cells (*Shi et al., 2015*; *Zhu et al., 2014*). These data suggested that ACKR4 might regulate

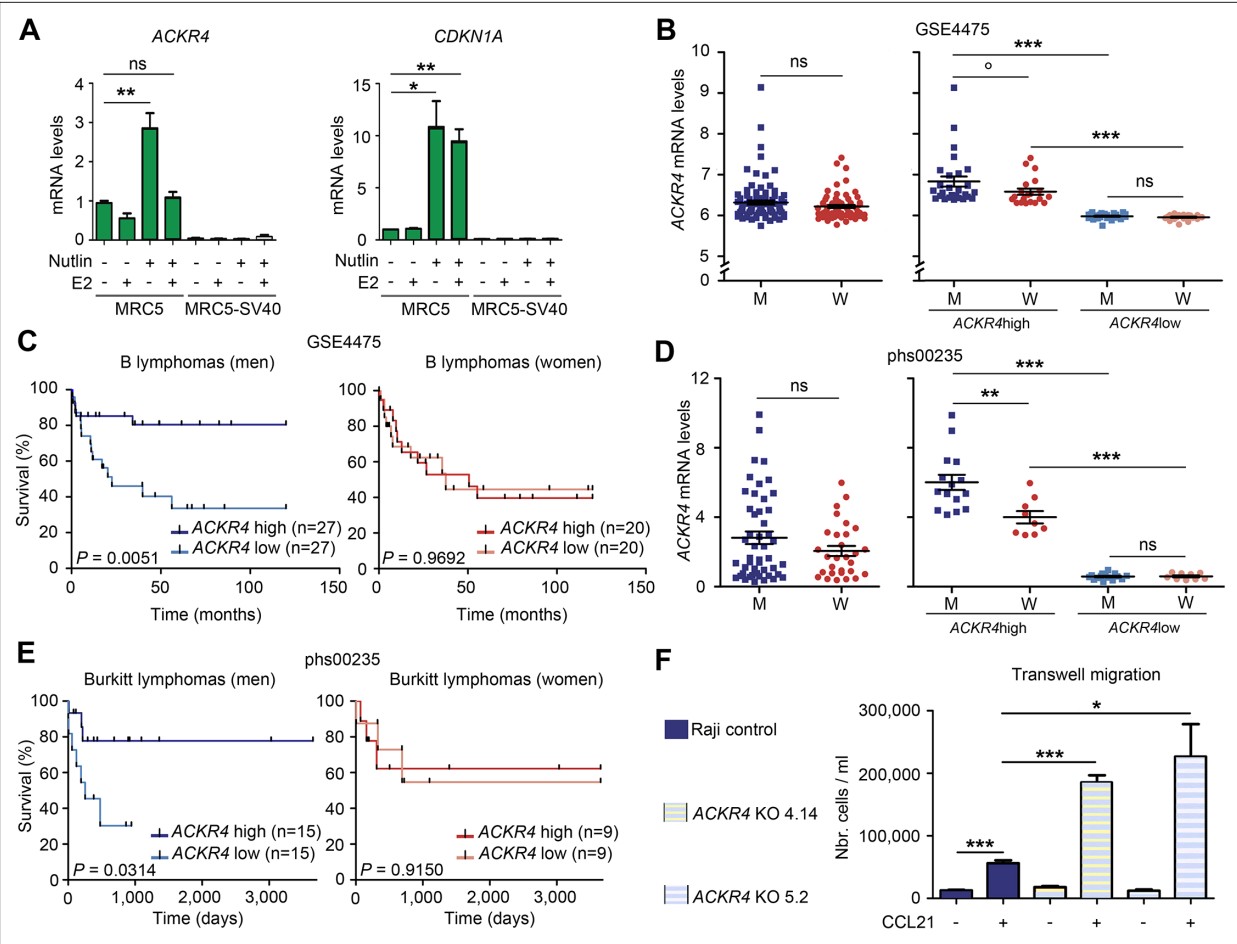

**Figure 4.** *ACKR4* is a male-specific prognostic factor in Burkitt lymphoma. (**A**) In human cells, p53 activation leads to an increased *ACKR4* expression abrogated by estradiol. *ACKR4* and *CDKN1A* mRNAs were quantified by RT-qPCR from p53-proficient (MRC5) and p53-deficient (MRC5-SV40) human fibroblasts, untreated or treated with Nutlin and/or estradiol (E2). Means ± SEM from four independent experiments. (**B, C**) Analysis of lymphoma dataset #GSE4475. *ACKR4* gene expression was plotted for all lymphoma patients with clinical follow-up (91 men [M], 68 women [W]), classified according to sex (**B**, left). Gene expression (**B**, right) or survival curves (**C**) were plotted for the 30% patients (27 men, 20 women) with the highest or lowest *ACKR4* expression, classified according to sex. (**D, E**) Analysis of Burkitt lymphoma-specific dataset #phs00235. *ACKR4* gene expression was plotted for all patients with a Burkitt lymphoma diagnosed at age 0–17 (48 males, 29 females), classified according to sex (**D**, left). Gene expression (**D**, right) or survival curves (**E**) were plotted for the 30% patients (15 men, 9 women) with the highest or lowest *ACKR4* expression, classified according to sex. (**F**) The knockout of *ACKR4* in Burkitt lymphoma Raji cells increases their CCL21-guided migration. Chemotaxis was assayed by using Boyden chambers with bare polycarbonate membranes as previously described (*Calpe et al., 2011*). Equal number of cells were deposited on the membrane of a transwell insert, then migration was determined by counting cells in the lower compartment, after 15 hr of culture with or without CCL21 added to the lower chamber. Statistical analyses by Student's *t* or Mann–Whitney tests (**A, B, D, F**) and Mantel–Cox (**C, E**) test. ***p<0.001, **p<0.01, *p<0.01, °p=0.054, ns: nonsignificant.

The online version of this article includes the following figure supplement(s) for figure 4:

**Figure supplement 1.** Further analyses of human B-cell lymphoma or multiple myeloma datasets.

**Figure supplement 2.** Strategy to knockout *ACKR4* in Burkitt lymphoma cells.

**Figure supplement 3.** Characterization of *ACKR4* KO Burkitt lymphoma cell clones.

**Figure supplement 4.** The knockout of *ACKR4* in Burkitt lymphoma Raji cells does not impact their proliferation.

the behavior of Burkitt lymphoma cells. To test this hypothesis, we designed a CRISPR-Cas9 approach to perform the knockout of *ACKR4* in Raji cells. Raji is a Burkitt lymphoma cell line isolated from a 11-year-old male patient (*Pulvertaft, 1964*). Although p53 is mutated in Raji cells (*Duthu et al., 1992*), these cells overexpress MYC (*Hamlyn and Rabbitts, 1983*) and express both ACKR4 and CCR7 (*Ferreira et al., 2014*), suggesting that they might be suitable to evaluate the impact of ACKR4 on Burkitt lymphoma cell behavior, particularly their migratory capacities. Raji cells were transfected

with a vector expressing Cas9, a puromycin resistance gene and either of two guide RNAs targeting *ACKR4* (or no guide RNA for control), then puromycin-resistant cells were selected and recovered as cellular pools or diluted to isolate cellular clones (*Figure 4—figure supplement 2*). Both guide RNAs targeted sequences in *ACKR4* mapping upstream an encoded DRY motif essential for signal transduction (*Watts et al., 2013*), so that Cas9-induced DNA breaks would generate knockout alleles. With this strategy, we obtained puromycin-resistant clones from Raji cells, two of which were verified to be *ACKR4* KO clones by DNA sequencing (*Figure 4—figure supplement 3*). We compared the proliferation and migration capacities of Raji control cells (transfected with the Cas9 expression vector without guide RNAs) and the two independent *ACKR4* KO Raji clones identified, cultured for 15 hr in medium supplemented or not with the CCL21 chemokine. Under these conditions, Raji cells of all genotypes appeared to proliferate similarly (*Figure 4—figure supplement 4*). Strikingly however, the KO of *ACKR4* led to a fourfold increase in CCL21-mediated cell migration, consistent with the hypothesis that ACKR4 may hinder MYC-driven B-cell lymphomagenesis (*Figure 4F*).

## Discussion

Here, we analyzed a mouse model with a specific deletion of the AS exon of the *Trp53* gene. Despite a subtle phenotype, this model revealed that a male-specific protective effect against *Eµ-Myc*-induced B-cell lymphomas is lost in the absence of p53-AS isoforms. $Trp53^{\Delta AS/\Delta AS}$ males also appeared more prone to develop spontaneous lymphomas, suggesting that the sex-specific protective effect conferred by p53-AS isoforms might not be restricted to the *Eµ-Myc* model.

Our transcriptomic data from splenic cells of $Trp53^{+/+}$ *Eµ-Myc* and $Trp53^{\Delta AS/\Delta AS}$ *Eµ-Myc* males disclosed very few differentially expressed genes and highlighted *Ackr4* as a male-specific positive prognostic factor in *Eµ-Myc*-induced lymphomas. Mechanistically, we identified *Ackr4*, expressed at low levels in splenic cells, as a p53 target gene that may be transactivated by p53-α and/or p53-AS according to luciferase assays. That *Ackr4* might be regulated by both types of p53 isoforms was expected because their DNA binding domains are identical. In fact, if one considers that p53-α isoforms appear more abundant than p53-AS isoforms in wild-type cells, and that the loss of p53-AS isoforms correlated with a decrease in p53-α levels in the thymocytes and tumor lymph nodes of mutant mice, then it seems likely that the reduced transactivation of *Ackr4* in the splenic cells of $Trp53^{\Delta AS/\Delta AS}$ *Eµ-Myc* males could mainly result from decreased p53-α levels, rather than the loss of p53-AS isoforms per se. In addition, we observed that 17-β estradiol can inhibit the p53-mediated transactivation of *Ackr4*. Together, our data suggest that *Ackr4* may be regulated by p53-α, p53-AS, and estrogens, likely accounting for sex-specific and p53-status-dependent differences in gene expression.

Our analyses reveal that in both mice and humans *Ackr4/ACKR4* is a male-specific prognostic factor in Burkitt-like lymphomas. Furthermore, several lines of evidence suggest that Ackr4 might act as a tumor suppressor of Myc-driven B-cell lymphomas. As mentioned before, the ACKR4-mediated sequestration of CCL21 may impair the CCR7 signaling cascade, which might lead to decreased MYC activity (*Shi et al., 2015*). In $Trp53^{+/+}$ *Eµ-Myc* male splenic cells, the observed lower expression of *Mt2*, known to be regulated by Myc (*Qin et al., 2021*), and of many genes that are hallmark Myc targets (as revealed by GSEA), appear consistent with this hypothesis. In addition, Ccr7 is required for lymphoma cell lodging to secondary lymphoid organs (*Rehm et al., 2011*) and ACKR4 expression was inversely correlated with the metastasis capacity of different types of cancer cells (*Shi et al., 2015*; *Zhu et al., 2014*). Consistent with this, we found that the KO of *ACKR4* in Raji Burkitt lymphoma cells led to a dramatic increase in CCL21-guided cell migration. Finally, Ackr4 regulates B cell differentiation (*Kara et al., 2018*), which raises the possibility that an alteration of the p53-Ackr4 pathway in $Trp53^{\Delta AS/\Delta AS}$ *Eµ-Myc* male splenic cells might contribute to increase the pools of pre-B and immature B cells that may be prone to lymphomagenesis. In sum, a decrease in Ackr4 expression might promote B-cell lymphomagenesis through several non-exclusive mechanisms. Importantly, ACKR4 was previously found to inhibit the growth and metastasis of breast, cervical, colorectal, hepatocellular, and nasopharyngeal cancer cells (*Feng et al., 2009*; *Hou et al., 2013*; *Ju et al., 2019*; *Shi et al., 2015*; *Zhu et al., 2014*), although no report mentioned any sex-specific bias for cancers occurring in both sexes. Our data provide evidence that sex-specific differences in *Ackr4* expression may have prognostic value. This suggests that measuring *ACKR4* gene expression in male patients with Burkitt lymphoma could be useful to identify the patients at higher risk, for whom specific therapeutic regimens might be required.

Interestingly, our data suggested that *Mt2* might be a male-specific negative prognostic factor in murine *Eµ-Myc*-induced lymphomas, but *MT2A* expression levels had no prognostic value in human lymphomas. A possible explanation for this discrepancy is suggested by the fact that *Mt2* is regulated by Myc. A translocation leading to MYC overexpression drives oncogenesis in all Burkitt lymphomas, but half of them exhibit additional missense MYC mutations enhancing its tumorigenicity (*Chakraborty et al., 2015*). The transcriptional program of a WT and two lymphoma-associated Myc mutants were recently compared, and we noticed that one of the mutants led to an alteration in *Mt2* expression (*Mahani et al., 2021*), which would abrogate any potential prognostic value.

Finally, a polymorphism in the *MDM2* gene promoter provided evidence that sex-specific hormones may affect p53 signaling and tumorigenesis (*Bond and Levine, 2007*). More recently, a higher frequency of *TP53* mutations in men, together with an increased vulnerability to alterations of X-linked genes encoding p53 regulators, was proposed to explain a higher cancer incidence and death in male patients (*Haupt et al., 2019*). Here, on the contrary, male mice and a subset of male patients were more efficiently protected against Burkitt-like lymphomas, which adds another layer of complexity to sex-specific differences in tumorigenesis. The p53 pathway thus underlies cancer sex-disparities through multiple mechanisms, which may notably include variations in p53 isoforms or Ackr4 expression.

# Materials and methods

**Key resources table**

| Reagent type (species) or resource | Designation | Source or reference | Identifiers | Additional information |
|---|---|---|---|---|
| Gene (*Mus musculus*) | *Trp53* | GenBank | ENSMUSG00000059552 | |
| Gene (*M. musculus*) | *Ackr4* | GenBank | ENSMUSG00000079355 | |
| Gene (*M. musculus*) | *Mt2* | GenBank | ENSMUSG00000031762 | |
| Gene (*Homo sapiens*) | *ACKR4* | GenBank | ENSMUSG00000129048 | |
| Strain, strain background (*M. musculus*, both sexes) | $Trp53^{\Delta AS}$, C57Bl/6J | *Simeonova et al., 2013* | | |
| Strain, strain background (*M. musculus*, both sexes) | *Eµ−Myc*, C57Bl/6J | Jackson Labs | B6.Cg-Tg(IghMyc)22Bri/J | |
| Strain, strain background (*M. musculus*, females) | CD-1 Nude CD-1 | Charles River Labs | Crl:CD1-$Foxn1^{nu}$ | |
| Strain, strain background (*M. musculus*, both sexes) | C57Bl/6J | Charles River Labs | | |
| Cell line (*M. musculus*, both sexes) | WT, $Trp53^{+/\Delta AS}$, $Trp53^{\Delta AS/\Delta AS}$, $Trp53^{+/-}$, $Trp53^{-/-}$ fibroblasts | This paper | Primary fibroblasts prepared from E13.5 days embryos | 'Cells and cell culture reagents' |
| Cell line (*H. sapiens*, male) | MRC5 | Sigma-Aldrich | MRC5 PD19 (#05072101) | |
| Cell line (*H. sapiens*, male) | MRC5-SV40 | Sigma-Aldrich | MRC5-SV2 (#84100401) | |
| Cell line (*H. sapiens*, female) | HEK293T | ATCC | CRL-3216 | |
| Cell line (*H. sapiens*, male) | Raji | ATCC | CCL-86 | |

*Continued on next page*

*Continued*

| Reagent type (species) or resource | Designation | Source or reference | Identifiers | Additional information |
|---|---|---|---|---|
| Cell line (*H. sapiens*, male) | Raji *ACKR4* KO | This paper | *ACKR4* KO 4.14 & 5.2 Raji derivatives | *Figure 4—figure supplement 3* |
| Transfected construct (Adenoviral E1A) | pWZL-E1A12S | Addgene | pWZL hygro 12S E1A (#18748) | |
| Transfected construct (human Ras) | pBabe-Hrasv12 | Addgene | pBabe-puro Ras v12 (# 1768) | |
| Antibody | p53 (rabbit polyclonal) | Novocastra | Leica NCL-p53-CM5p | 1/2000 |
| Antibody | Myc (mouse monoclonal) | Santa Cruz | 9E-10 sc40 | 1/1000 |
| Antibody | p21 (mouse monoclonal) | Santa Cruz | F-5 sc6246 | 1/200 |
| Antibody | Actin (mouse monoclonal) | Santa Cruz | Actin-HRP sc47778 | 1/5000 |
| Antibody | CD45R/B220 APC (rat, monoclonal) | BD Biosciences | Anti-mouse CD45R/B220 APC (#561880) | 1/200 |
| Antibody | IgD (rat, monoclonal) | BD Biosciences | Anti-mouse IgD BV 605 (#563003) | 1/100 |
| Antibody | CD43 (rat, monoclonal) | BD Biosciences | Anti-mouse CD43 FITC (#561856) | 1/200 |
| Antibody | IgM (rat, monoclonal) | BD Biosciences | Anti-mouse IgM PE (#562033) | 1/50 |
| Recombinant DNA reagent | pSpCas9(BB)–2A-Puro | Addgene | PX459 (#48139) | |
| Sequence-based reagent | Trp53α-F | This paper | qPCR primer | AAAGGATGCCCATGCTACAGA; *Figure 1—figure supplement 1* |
| Sequence-based reagent | Trp53α-R | This paper | qPCR primer | TCTTGGTCTTCAGGTAGCTGGAG; *Figure 1—figure supplement 1* |
| Sequence-based reagent | Trp53AS-F | This paper | qPCR primer | AAAGGATGCCCATGCTACAGA; *Figure 1—figure supplement 1* |
| Sequence-based reagent | Trp53AS-R | This paper | qPCR primer | TGAAGTGATGGGAGCTAGCAGTT; *Figure 1—figure supplement 1* |
| Sequence-based reagent | Ackr4-F | This paper | qPCR primer | GCACCTCTCCCAGCTTAAACA; *Figure 3* |
| Sequence-based reagent | Ackr4-R | This paper | qPCR primer | AATAGTATTCCGCTGACTGGTTCAG; *Figure 3* |
| Sequence-based reagent | ACKR4-F | This paper | qPCR primer | ACTGCTCCTCTCTGCCGACTAC; *Figure 4* |
| Sequence-based reagent | ACKR4-R | This paper | qPCR primer | GCCATTCATTTCATTTTCCTCAT; *Figure 4* |
| Sequence-based reagent | ACKR4-g4 | This paper | Guide for CRISPR #4 | TGGTAGTGGCAATTTATGCC; *Figure 4—figure supplement 3* |
| Sequence-based reagent | ACKR4-g5 | This paper | Guide for CRISPR #5 | GGGCTGTTAATGCAGTTCAT; *Figure 4—figure supplement 3* |
| Peptide, recombinant protein | CCL21 | Preprotech | #300-35A | |
| Peptide, recombinant protein | Superscript IV | Invitrogen | TF #18090010 | |
| Commercial assay or kit | Nucleospin RNA II | Macherey-Nagel | FS #NZ74095520 | |

*Continued on next page*

*Continued*

| Reagent type (species) or resource | Designation | Source or reference | Identifiers | Additional information |
|---|---|---|---|---|
| Commercial assay or kit | Power SYBR Green | Applied Biosystems | # 4367659 | |
| Commercial assay or kit | Supersignal West Femto | Thermo Fisher | # 34096 | |
| Commercial assay or kit | AnnexinV-FITC apoptosis staining/ detection kit | Abcam | # Ab14085 | |
| Commercial assay or kit | Truseq stranded Total RNA | Illumina | #20020596 | |
| Commercial assay or kit | Nucleofector Amaxa kit V | Lonza | # VCA-1003 | |
| Chemical compound, drug | Doxorubicin | Sigma-Aldrich | # D1515 | |
| Chemical compound, drug | Etoposide | Sigma-Aldrich | # E1383 | |
| Chemical compound, drug | Nutlin 3a | Sigma-Aldrich | # SML-0580 | |
| Chemical compound, drug | 17β-estradiol | Sigma-Aldrich | # E2758 | |
| Software, algorithm | FlowJo | Beckton-Dickinson | RRID:SCR_008520 | v 10.10 |
| Software, algorithm | featureCounts | *Liao et al., 2014* | | |
| Software, algorithm | DESeq2 R package | *Love et al., 2014* | | |
| Software, algorithm | GSEA software | *Subramanian et al., 2005* | | |
| Software, algorithm | PWMScan | *Ambrosini et al., 2018* | | |
| Software, algorithm | CRISPOR | *Haeussler et al., 2016* | | |
| Software, algorithm | Prism | GraphPad | RRID:SCR_002798 | v 5.0 |

## Mice

Design and construction of the *Trp53^{ΔAS}* mouse model were previously described (*Simeonova et al., 2013*). A minimum of 10 backcrosses with C57Bl/6J mice of both sexes (Charles River Laboratories) were performed before establishing the cohorts of *Trp53^{+/+}* and *Trp53^{ΔAS/ΔAS}* littermate mice used in this study. Mouse genotyping with multiple primer sets confirmed >99% C57Bl/6J genetic background after 10 backcrosses (primer sequences available upon request). Cohorts of *Trp53^{+/+} Eμ-Myc* and *Trp53^{ΔAS/ΔAS} Eμ-Myc* mice were established with identical parental origin of the *Eμ-Myc* transgene. For all experiments, mice housing and treatment were conducted according to the Institutional Animal Care and Use Committee of the Institut Curie (approved project #03769.02).

## Cells and cell culture reagents

MEFs were isolated from 13.5 days post-coitum embryos and cultured in a 5% $CO_2$ and 3% $O_2$ incubator, in Dulbecco's Modified Eagle Medium (DMEM) GlutaMAX (Gibco), with 15% Fetal Bovine Serum (FBS) (PAN Biotech), 100 μM 2-mercaptoethanol (Millipore), 0.1 mM non-essential amino acids and penicillin/streptomycin (Gibco) for less than five passages, except for 3T3 experiments, performed in a 5% $CO_2$ incubator for nine passages. Cells were treated for 24 hr with 0.5 μg/ml doxorubicin (Sigma-Aldrich), 15 μM etoposide (Sigma-Aldrich), 10 μM Nutlin 3a (*Vassilev et al., 2004*) (Sigma-Aldrich), and/or 5 μg/ml 17β-estradiol (Sigma-Aldrich). At least three independent experiments with at least two independent littermate MEF clones of each genotype and each sex were performed to measure

DNA damage responses. For estradiol assays, four independent experiments with three independent MEF male clones of each genotype were performed. Human lung fibroblast MRC5 and its SV40-transformed derivatives were cultured in a 5% $CO_2$ and 3% $O_2$-regulated incubator in Minimum Essential Medium (MEM) medium without Phenol Red (Gibco), completed with 10% FBS, 2 mM L-glutamine (Gibco), 1 mM pyruvate, 0.1 mM non-essential amino acids, and penicillin/streptomycin, and treated for 24 hr with 10 µM Nutlin 3a and/or 5 µg/ml 17β-estradiol (Merck). Four independent experiments were performed. Burkitt lymphoma cells (Raji or Raji *ACKR4* KO derivatives) were cultured in a 5% $CO_2$ incubator in RPMI GlutaMAX (Gibco) supplemented with 10% FBS, 2 mM L-glutamine, 1 mM pyruvate, 0.1 mM non-essential amino acids, 0.45% glucose (Sigma-Aldrich), and penicillin/streptomycin. In proliferation and migration assays, Raji control cells and two independent *ACKR4* KO derivatives were treated or not for 15 hr with 1 µg/ml CCL21 (PreproTech) then counted in triplicates. For all cell lines, cell culture supernatants were found negative for mycoplasma contamination.

## Quantitative RT-PCR

Total RNAs were extracted using nucleospin RNA II (Macherey-Nagel), reverse-transcribed using superscript IV (Invitrogen), and real-time quantitative PCRs were performed on an ABI PRISM 7500 using Power SYBR Green (Applied Biosystems) as previously described (*Simeonova et al., 2012*). For quantification of p53 isoforms in healthy tissues, a forward primer in exon 10 and a reverse primer encompassing the boundary between exons 10 and 11 were used for p53-α amplification, whereas the same forward primer and a reverse primer located in exon AS were used for p53-AS amplification (see *Supplementary file 3* for primer sequences). To determine the AS/α mRNA ratios, expression levels were compared with a standard curve generated by serial dilutions of a plasmid containing both p53-AS and p53-α cDNAs.

## Western blots

Thymocytes were lysed in RIPA buffer (50 mM Tris–HCl pH 8, 150 mM NaCl, 5 mM EDTA, 0.5% deoxycholic acid, 0.1% SDS, 1% NP-40) with a cocktail of protease inhibitors (Roche) and 1 mM PMSF (Sigma). Whole-cell extracts were sonicated three times for 10 s and centrifuged at 13,000 r.p.m. for 30 min to remove cell debris. MEFs or B-cell lymphomas were lysed in Giordano's buffer (50 mM Tris–HCl pH 7.4, 250 mM NaCl, 5 mM EDTA, 0.1% Triton X-100) with a cocktail of protease inhibitors (Roche) and 1 mM PMSF (Sigma). Protein lysate concentration was determined by bicinchoninic acid (BCA) assay (Thermo Scientific) and 30 µg of each lysate was fractionated by SDS–PAGE on a 4–12% polyacrylamide gel and transferred onto polyvinylidene difluoride (PVDF) membrane (Amersham). Membranes were incubated with antibodies against p53 (CM5, Novocastra), myc (9E-10, Santa Cruz), p21 (F-5, Santa Cruz), and actin (actin-HRP sc47778, Santa Cruz) and revealed with SuperSignal West femto detection reagent (Thermo Scientific).

## Apoptosis assays

Six-week-old *Trp53*[+/+] and *Trp53*[ΔAS/ΔAS] male mice were whole-body irradiated with 5 Gy of γ-irradiation. Mice were sacrificed 4 hr later and thymocytes were recovered, stained with AnnexinV-FITC Apoptosis detection kit (Abcam), then analyzed by flow cytometry using FlowJo.

## Cell-cycle assays

Log-phase MEFs were irradiated at room temperature with a CS γ-irradiator at doses of 3 or 10 Gy, incubated for 24 hr, then pulse-labeled for 1 hr with 10 µM BrdU, fixed in 70% ethanol, double-stained with FITC anti BrdU and propidium iodide, and sorted by flow cytometry using a BD FACSort. Data were analyzed using FlowJo.

## Oncogene-induced tumor xenografts

MEFs with the indicated genotypes were sequentially infected with pWZL-E1A12S and pBABE-Hrasv12 viruses as previously described (*Toledo et al., 2006*). In total, $5 \times 10^6$ E1A- and Ras- (E1A+Ras) expressing MEFs of each genotype were injected subcutaneously into the flanks of 7-week-old female athymic nude mice (at least four mice per genotype) and tumor volumes were determined 1, 8, 15, 21, and 25 days after injection. Importantly, populations of (E1A+Ras)-expressing cells were used to minimize potential differences in expression levels that could result from independent viral insertion sites.

## Cell sorting of B-cell subpopulations

Splenic cells were recovered from 6-week-old asymptomatic mice and incubated with DAPI and the following antibodies: APC rat anti-mouse CD45R/B220, FITC rat anti-mouse CD43, PE rat anti-mouse IgM, and BV605 rat anti-mouse IgD (BD Pharmingen). First, the B220+ CD43 cells were selected by flow cytometry from DAPI-negative living cells, yielding subsequently four different B subpopulations based on IgM and IgD labeling: IgM-/IgD- preB lymphocytes, IgM low/IgD- immature B lymphocytes, IgM high/IgD- transitional B lymphocytes, and IgM+/IgD+ mature B lymphocytes.

## RNA-seq analysis

Total RNA was extracted from the spleen of 4–6-week--old asymptomatic mice using nucleospin RNA II (Macherey-Nagel). The quality of RNA was checked with Bioanalyzer Agilent 2100 and RNAs with a RIN (RNA integrity number) > 6 were retained for further analysis. RNA was depleted from ribosomal RNA, then converted into cDNA libraries using a TruSeq Stranded Total Library preparation kit (Illumina). Paired-end sequencing was performed on an Illumina MiSeq platform. Reads were mapped to the mouse genome version GRCm38 and counted on gene annotation gencode.vM18 with feature-Counts (*Liao et al., 2014*). Differentially expressed genes of C57Bl/6J genetic background with an adjusted p-value<0.05 were identified using the DESeq2 R package (*Love et al., 2014*). Gene set enrichment analysis was performed using the GSEA software with canonical pathway gene sets from the Mouse Molecular Signature Database (*Subramanian et al., 2005*).

## Luciferase assays

The candidate p53 responsive element (p53 RE) in the *Ackr4* promoter was identified using the JASPAR database of binding profiles (*Fornes et al., 2020*) with the position weight matrix scanner PWMscan (*Ambrosini et al., 2018*). A 1.5 kb fragment from *Ackr4* intron 1, containing a WT or mutant p53 RE at its center, was cloned upstream an SV40 minimal promoter and a luciferase reporter gene in the backbone of a PGL3 plasmid (Promega). We used lipofectamine 2000 to transfect $Trp53^{-/-}$ MEFs with 2 µg of either luciferase expression vector, 2 µg of an expression vector for $p53^{WT}$, $p53^{AS}$ or the DNA-binding mutant $p53^{R270H}$, and 30 ng of a renilla luciferase expression plasmid (pGL4.73, Promega) for normalization. Transfected cells were incubated for 24 hr, then trypsinized, resuspended in 75 µl culture medium with 7.5% FBS, and transferred into a well of an optical 96-well plate (Nunc). The dual-glo luciferase assay system (Promega) was used according to the manufacturer's protocol to lyse the cells and read firefly and renilla luciferase signals. Results were normalized, then the average luciferase activity in cells transfected with the WT p53RE luciferase reporter and the $p53^{R270H}$ expression plasmid was assigned a value of 1.

## Generation of *ACKR4* knockout Burkitt lymphoma cells

Six guide RNAs (gRNAs) were designed to target the human *ACKR4* gene using the web tool CRISPOR (*Haeussler et al., 2016*). For each gRNA, two reverse complementary oligonucleotides were designed (with added backbone sequences, including a 5' G nucleotide for gRNA sequences without a 5' G to improve transcription efficiency from a U6 promoter), annealed, and cloned in the Cas9 expression vector pSpCas9(BB)–2A-Puro (PX459) (Addgene). Preliminary tests for efficiency to induce cleavage in the targeted DNA regions were performed in HEK293T cells, which led to select the gRNAs #4 (5'-TGGTAGTGGCAATTTATGCC-3') and #5 (5'-GGGCTGTTAATGCAGTTCAT-3') for further experiments. Raji Burkitt lymphoma cells were transfected with a PX459 vector expressing gRNA #4 or #5 (or no gRNA for control) using Nucleofector Amaxa kit V (Lonza), and selection was carried out with 1 µg/ml puromycin for 3 weeks to obtain stably transfected cells. Clones were obtained by seeding cells at low density (1 cell/5 wells) in 96-well plates, expanded, then subdivided into two parts, half for storage in liquid nitrogen and half for *ACKR4* genotyping by Sanger DNA sequencing (see *Supplementary file 3* for primer sequences). For each cell clone, the *ACKR4* target DNA regions were amplified by PCR, PCR products were cloned in a PGL3 plasmid, and eight transformed bacteria colonies were sequenced (*Figure 4—figure supplement 2*). This led to identify two homozygous *ACKR4* KO clones, one (clone 4.14) obtained using gRNA #4 and one (clone 5.2) using gRNA #5 (*Figure 4—figure supplement 3*).

## Cell migration assay

Cell migration assays toward the chemokine CCL21 were performed with Boyden chambers essentially as described (*Calpe et al., 2011*) by measuring transwell migration across bare polycarbonate

membranes with a pore size of 5 µm (Corning). A total of 100 µl of culture medium (RPMI GlutaMAX, 10% FBS, 2 mM L-glutamine, 1 mM pyruvate, 0.1 mM non-essential amino acids, 0.45% glucose, and penicillin/streptomycin) containing $5 \times 10^5$ cells was added to a 6.5 mm diameter transwell insert, and 600 µl of culture medium with or without 1 µg/ml CCL21 were added to the lower compartment. After 15 hr at 37°C in 5% $CO_2$, the number of migrated cells in the lower chamber was determined using a Coulter Counter.

## Statistical analyses

Student's unpaired *t*-tests were used in most figures to analyze the differences between WT vs. ΔAS values. Log-rank (Mantel–Cox) tests were used to analyze Kaplan–Meier tumor-free survival curves. Analyses were performed using GraphPad Prism 5, and values of $p<0.05$ were considered significant.

## Acknowledgements

This project was supported by grants from the Comité Tumeurs of Fondation de France (to FT), the Comité Ile-de-France and Comité national (Labellisation 2014–18) of the Ligue Nationale Contre le Cancer (to FT), and the Fondation ARC pour la recherche sur le Cancer (to FT). IS, JR, and EE were PhD fellows of the Ministère de la Recherche; JL and AMorin were postdoctoral fellows of Cancéropôle Ile-de-France and Institut National du Cancer. MG was paid by European the Research Council 875532-Prostator-ERC-2019-PoC attributed to AMoril; JCB was supported by Cancer Research-UK. We thank K Fernandes for his participation in making the ΔAS mutation, and members of the Institut Curie platforms: I Grandjean, H Gautier, C Daviaud, M Garcia, M Verlhac, A Fosse, and P Bureau (animal facility); S Baulande and S Lameiras (NGS); M Huerre, A Nicolas, and R Leclere (histopathology); Z Maciorowski, A Viguier, and S Grondin (flow cytometry).

## Additional information

### Funding

| Funder | Grant reference number | Author |
|---|---|---|
| Fondation de France | Comité tumeurs | Franck Toledo |
| Ligue Contre le Cancer | Labellisation | Franck Toledo |
| Ligue Contre le Cancer | Comité Ile de France | Franck Toledo |
| Fondation ARC pour la Recherche sur le Cancer | | Franck Toledo |
| European Research Council | 875532-Prostator-ERC-2019-PoC | Antonin Morillon |

The funders had no role in study design, data collection and interpretation, or the decision to submit the work for publication.

### Author contributions

Anne Fajac, Conceptualization, Formal analysis, Investigation, Visualization, Writing – original draft, Writing – review and editing; Iva Simeonova, Investigation, Initiated the project; Julia Leemput, Aurélie Morin, Vincent Lejour, Annaïg Hamon, Jeanne Rakotopare, Wilhelm Vaysse-Zinkhöfer, Eliana Eldawra, Investigation; Marc Gabriel, Marina Pinskaya, Antonin Morillon, Formal analysis; Jean-Christophe Bourdon, Resources, Initiated the project; Boris Bardot, Conceptualization, Formal analysis, Supervision, Investigation, Visualization, Writing – original draft, Writing – review and editing; Franck Toledo, Conceptualization, Formal analysis, Supervision, Funding acquisition, Investigation, Visualization, Writing – original draft, Project administration, Writing – review and editing, Initiated the project

### Author ORCIDs

Vincent Lejour 
Antonin Morillon 
Jean-Christophe Bourdon 

Boris Bardot ⓘ https://orcid.org/0000-0003-4976-9593
Franck Toledo ⓘ https://orcid.org/0000-0003-3798-4106

### Ethics

For all experiments, mice housing and treatment were conducted according to Institutional Animal Care and Use Committee of the Institut Curie (approved project #03769.02).

Reviewer #1 (Public review): https://doi.org/10.7554/eLife.92774.3.sa1
Reviewer #2 (Public review): https://doi.org/10.7554/eLife.92774.3.sa2
Author response https://doi.org/10.7554/eLife.92774.3.sa3

## Additional files

### Supplementary files

• Supplementary file 1. Expression of *Ackr4, Cdkn1a,* and *Mdm2* in *Trp53$^{+/+}$ Eμ-Myc* and *Trp53$^{ΔAS/ΔAS}$ Eμ-Myc* male splenic cells. Read numbers for the indicated genes, obtained by Bulk RNA-seq from the spleens of three *Trp53$^{+/+}$ Eμ-Myc* (WT_Myc) and four *Trp53$^{ΔAS/ΔAS}$ Eμ-Myc* (ΔAS_Myc) male mice.

• Supplementary file 2. Details of the gene set enrichment analysis (GSEA) for hallmark Myc targets. Datasets from *Trp53$^{+/+}$ Eμ-Myc* and *Trp53$^{ΔAS/ΔAS}$ Eμ-Myc* male splenic cells were analyzed by GSEA. A normalized enrichment score of 2.4965038 was found for the gene set 'Hallmark_Myc_targets_V1' in *Trp53$^{ΔAS/ΔAS}$ Eμ-Myc* males. The table provides details on the profile represented in *Figure 3K*, with scores and positions of gene set members on the rank ordered list.

• Supplementary file 3. Oligonucleotide sequences.

• MDAR checklist

### Data availability

RNA sequencing data have been deposited in the Gene Expression Omnibus (GEO) under the accession code GSE209708. All other data are available within the article and its supplementary information.

The following dataset was generated:

| Author(s) | Year | Dataset title | Dataset URL | Database and Identifier |
|---|---|---|---|---|
| Fajac A, Bardot B, Toledo F, Gabriel M | 2024 | Splenocyte mRNA profiles of 4-6 weeks-old p53+/+ Em-Myc and p53DAS/DAS Em-Myc mice | https://www.ncbi.nlm.nih.gov/geo/query/acc.cgi?acc=GSE209708 | NCBI Gene Expression Omnibus, GSE209708 |

The following previously published datasets were used:

| Author(s) | Year | Dataset title | Dataset URL | Database and Identifier |
|---|---|---|---|---|
| Hummel M, Bentink S, Berger H, Klapper W, Wessendorf S, Barth TF, Bernd H, Cogliatti S, Dierlamm J, Feller AC, Hansmann M, Haralambieva E, Harder L, Hasenclever D, Kuehn M, Lenze D, Lichter P, Martin-Subero JI, Moeller P, Mueller-Hermelink H, Ott G, Parwaresch RM, Pott C, Rosenwald A, Rosolowski M, Schwaenen C, Stuerzenhofecker B, Szczepanowski M, Trautmann H, Wacker H, Spang R, Loeffler M, Truemper L, Stein H, Siebert R | 2006 | A Biologic Definition of Burkitt's Lymphoma from Transcriptional and Genomic Profiling | https://www.ncbi.nlm.nih.gov/geo/query/acc.cgi?acc=GSE4475 | NCBI Gene Expression Omnibus, GSE4475 |
| Care MA, Barrens SL | 2021 | Whole genome expression profiling based on paraffin embedded tissue of a large DLBCL cohort | https://www.ncbi.nlm.nih.gov/geo/query/acc.cgi?acc=GSE181063 | NCBI Gene Expression Omnibus, GSE181063 |
| Danziger SA, McConnell M, Gockley J, Young MH, Rosenthal A, Schmitz F, Reiss DJ, Farmer P, Alapat DV, Singh A, Ashby C, Bauer M, Ren Y, Smith K, Couto SS, van Rhee F, Davies F, Zangari M, Petty N, Orlowski RZ, Dhodapkar M, Copeland W, Fox B, Hoering A, Fitch A, Newhall K, Barlogie B, Trotter MW, Hershberg RM, Walker BA, Dervan A, Ratushny AV, Morgan G | 2019 | Identifying a high-risk cellular signature in the multiple myeloma bone marrow microenvironment_2 | https://www.ncbi.nlm.nih.gov/geo/query/acc.cgi?acc=GSE136337 | NCBI Gene Expression Omnibus, GSE136337 |

*Continued on next page*

*Continued*

| Author(s) | Year | Dataset title | Dataset URL | Database and Identifier |
|---|---|---|---|---|
| Morin RD, Mendez-Lago M, Mungall AJ, Goya R, Mungall KL, Corbett RD, Johnson NA, Severson TM, Chiu R, Field M, Jackman S, Krzywinski M, Scott DW, Trinh DL, Tamura-Wells J, Li S, Firme MR, Rogic S, Griffith M, Chan S, Yakovenko O, Meyer IM, Zhao EY, Smailus D, Moksa M, Chittaranjan S, Rimsza L, Brooks-Wilson A, Spinelli JJ, Ben-Neriah S, Meissner B, Woolcock B, Boyle M, McDonald H, Tam A, Zhao Y, Delaney A, Zeng T, Tse K, Butterfield Y, Birol I, Holt R, Schein J, Horsman DE, Moore R, Jones SJ, Connors JM, Hirst M, Gascoyne RD, Marra MA | 2011 | National Cancer Institute Cancer Genome Characterization Initiative | https://www.ncbi.nlm.nih.gov/projects/gap/cgi-bin/study.cgi?study_id=phs000235.v21.p6 | NCBI dbGaP, phs000235.v21.p6 |

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
