## [Editor Report · eLife assessment]

This **important** study using engineered mouse models provides a first and **compelling** demonstration of a pathogenic phenotype associated with lack of expression of p53AS isoforms, isoforms of the p53 protein with a different C-terminus than canonical p53. The role of these isoforms has been elusive so far and this first demonstration represents a substantial advance in our understanding of the complex role(s) of p53 isoforms. The revised article adequately addresses previous concerns.

---

## [Referee Report · Reviewer #1 (Public review)]

Summary:

The authors originally investigated the function of p53 isoforms with an alternative C-terminus encoded by the Alternatively Spliced (AS) exon in place of exon 11 encoding the canonical "α" C-terminal domain. For this purpose, the authors create a mouse model with a specific deletion of the AS exon.

Strengths:

Interestingly, wt or p53ΔAS/ΔAS mouse embryonic fibroblasts did not differ in cell cycle control, expression of well-known p53 target genes, proliferation under hyperoxic conditions, or the growth of tumor xenografts. However, p53-AS isoforms were shown to confer male-specific protection against lymphomagenesis in Eμ-Myc transgenic mice, prone to highly penetrant B-cell lymphomas. In fact, p53ΔAS/ΔAS Eμ-Myc mice were less protected from developing B-cell lymphomas compared to WT counterparts. The important difference that the authors find between WT and p53ΔAS/ΔAS Eμ-Myc males is a higher number of immature B cells in p53ΔAS/ΔAS vs WT mice. Higher expression of Ackr4 and lower expression of Mt2 was found in p53+/+ Eμ-Myc males compared to p53ΔAS/ΔAS counterparts, suggesting that these two transcripts are in part regulators of B-cell lymphomagenesis and enrichment for immature B cells.

The manuscript integrates an elegant genetic approach with in vivo analyses providing a robust set of data which strengthens the role of p53 isoforms in leukemogenesis.

---

## [Referee Report · Reviewer #2 (Public review)]

Summary:

This manuscript provides a detailed analysis of B-cell lymphomagenesis in mice lacking an alternative exon in region encoding the C-terminal (regulatory) domain of the p53 protein and thus enable to assemble the so-called p53AS isoform. This isoform differs from canonical p53 by the replacement of roughly 30 c-terminal residues by about 10 residues encoded by the alternative exon. There is biochemical and biological evidence that p53AS retains strong transcriptional and somewhat enhanced suppressive activities, with mouse models expressing protein constructs similar to p53AS showing signs of increased p53 activity leading to rapid and lethal anemia. However, the precise role of the alternative p53AS variant has not been addressed so far in a mouse model aimed at demonstrating whether the lack of this particular p53 isoform (trp53ΔAS/ΔAS mice) may cause a specific pathological phenotype.

Results show that lack of AS expression does not noticeably affect p53 the patterns of protein expression and transcriptional activity but reveals a subtle pathogenic phenotype, with trp53ΔAS/ΔAS males, but not females, tending to develop more frequently and earlier B-cell lymphoma than WT. Next, the authors then introduced ΔAS in transgenic Eμ-Myc mice that show accelerated lymphomagenesis. They show that lack of AS caused increased lethality and larger tumor lymph nodes in p53ΔAS Eμ-Myc males compared to their p53WT Eμ-Myc male counterparts, but not in females. Comparative transcriptomics identified a small set of candidate, differentially expressed gene, including Ackr4 (atypical chemokine receptor 4), which was significantly expressed in the spleens of ΔAS compared to WT controls. Ackr4 encodes a dummy receptor acting as an interceptor for multiple chemokines and thus may negatively regulate a chemokine/cytokine signalling axis involved in lymphomagenesis, which is down-regulated by estrogen signalling. Using in vitro cell models, the authors provide evidence that Ackr4 is a transcriptional target for p53 and that its p53-dependent activation is repressed by 17b-oestradiol. Finally, seeking evidence for a relevance for this gene in human lymphomagenesis, the authors analyse Burkitt lymphoma transcriptomic datasets and show that high ACKR4 expression correlated with better survival in males, but not in females

---

## [Author Response]

The following is the authors’ response to the original reviews.

**Reviewer #1 (Recommendations For The Authors):**
(1) In the first paragraph of the result section it is not clear why the authors introduce the function of p53ΔAS/ΔAS in thymocyte and then they mention fibroblasts. The authors should clarify this point. The authors should also explain based on what rationale they use doxorubicin and nutlin to analyze p53 activity (Figure 1 and figure S1).

We thank the reviewer for this comment. In the revised manuscript, we corrected this by mentioning, at the beginning of the Results section: “We analyzed cellular stress responses in thymocytes, known to undergo a p53-dependent apoptosis upon irradiation (Lowe et al., 1993), and in primary fibroblasts, known to undergo a p53-dependent cell cycle arrest in response to various stresses - e.g. DNA damage caused by irradiation or doxorubicin (Kastan et al., 1992), and the Nutlin-mediated inhibition of Mdm2, a negative regulator of p53 (Vassilev et al., 2004).”

(2) The authors should provide quantification for the western blot in figure 2D because the reduction of p53 protein level in mutant vs wt tumors is not striking.

In the previous version of the manuscript, the quantification of p53 bands had been included, but quantification results were mentioned below the actin bands, rather than the p53 bands, and this was probably confusing. We have corrected this in the revised version of the manuscript. The quantification results are now provided just below the p53 bands in Figs. 1B and 2D, which should clarify this point. For Figure 2D, the quantifications show a strong decrease in p53 levels for 3 out of 4 analyzed mutant tumors. For consistency purposes, in the revised manuscript the quantification results also appear below Myc bands in Fig. 2C.

(3) In the discussion section, the authors propose that a difference in Ackr4 expression may have prognostic value and that measuring ACKR4 gene expression in male patients with Burkitt lymphoma could be useful to identify the patients at higher risk. However the authors perform a lot of correlative analysis, both in mice and in patients, but the manuscript lacks of functional experiments that could help to functionally characterize Ackr4 and Mt2 in the etiology of B-cell lymphomas in males (both in mouse and in human models).

In the previous version of the manuscript, we proposed that Ackr4 might act as a suppressor of B-cell lymphomagenesis by attenuating Myc signaling. This hypothesis relied on studies showing that Ackr4 impairs the Ccr7 signaling cascade, which may lead to decreased Myc activity (Ulvmar et al., 2014; Shi et al., 2015; Bastow et al., 2021) and that the loss of Ccr7 may delay Myc-driven lymphomagenesis (Rehm et al., 2011). Furthermore, we proposed that the increased expression of Mt2 in p53ΔAS/ΔAS Em-Myc male splenic cells reflected an increase in Myc activity, because Mt2 is known to be regulated by Myc (Qin et al., 2021) and because the Mt2 promoter is bound by Myc in B cells according to experiments reported in the ChIP-Atlas database. However, in the first version of the manuscript this hypothesis might have appeared only partially supported by our data because an increase in Myc activity could be expected to have a more general impact, i.e. an impact not only on the expression of Mt2, but also on the expression of many canonical Myc target genes. In the revised manuscript, we show that this is indeed the case. We performed a gene set enrichment analysis (GSEA) comparing the RNAseq data from p53ΔAS/ΔAS Eμ-Myc and p53+/+ Eμ-Myc male splenic cells and found an enrichment of hallmark Myc targets in p53ΔAS/ΔAS Eμ-Myc cells. These new data, which strengthen our hypothesis of differences in Myc signaling intensity, are presented in Fig. 3K and Table S2.

Importantly, we now go beyond correlative analyses by providing direct experimental evidence that ACKR4 impacts on the behavior of Burkitt lymphoma cells. We used a CRISPR-Cas9 approach to knock-out ACKR4 in Raji Burkitt lymphoma cells and found that ACKR4 KO cells exhibited a 4-fold increase in chemokine-guided cell migration. These new data are presented in Figure 4F and the supplemental Figures S5-S7.

Finally, following a suggestion of Reviewer#2, we now also point out that “Ackr4 regulates B cell differentiation (Kara et al., 2018), which raises the possibility that an altered p53-Ackr4 pathway in p53ΔAS/ΔAS Eμ-Myc male splenic cells might contribute to increase the pools of pre-B and immature B cells that may be prone to lymphomagenesis.”

In sum, we now mention in the Discussion that a decrease in Ackr4 expression might promote B-cell lymphomagenesis through three non-exclusive mechanisms.

**Reviewer #2 (Recommendations For The Authors):**
(1) A great addition would be to demonstrate how p53AS specifically contributes to the regulation of Ackr4. In particular, is there evidence that p53AS might be preferentially recruited on p53 RE within that gene as compared to WT? The availability of specific antibodies that distinguish between AS and WT p53 might help to address this (experimentally complex) question. As a note, usage of such antibodies would also strengthen Fig 1B, in which the AS isoform appears as a mere faint shadow under p53, thus making its "disappearance" in trp53ΔAS/ΔAS difficult to evaluate.

We agree with the referee that efficient antibodies against p53-AS isoforms would have been useful. In fact, we tried a non-commercial antibody developed for that purpose, but it led to many unspecific bands in western blots and appeared not reliable. Importantly however, our luciferase assays clearly show that both p53-a and p53-AS can transactivate Ackr4, a result that might be expected because these isoforms share the same DNA binding domain. Furthermore, because p53-a isoforms appear more abundant than p53-AS isoforms at the protein and RNA levels (Figs. 1B and S1A), and because the loss of p53-AS isoforms leads to a significant decrease in p53-a protein levels (Figs. 1B and 2D), we think that in p53ΔAS/ΔAS cells the reduction in p53-a levels might be the main reason for a decreased transactivation of Ackr4. This is now more clearly discussed in the revised manuscript.

(2) A most interesting observation is in Fig3 A and Fig S3, showing that spleen cells of p53ΔAS Eμ-Myc males (but not females) were enriched in pre-B and immature B cells as compared to WT counterparts. This observation points to a possible defect in B cell maturation process. It would be most interesting to determine whether this particular defect is directly mediated by a p53AS-Ackr4 axis. The hypothesis raised by the authors in the Discussion section is that increased Ackr4 expression may delay lymphomatogenesis, but data in Fig 3A and 3S actually suggest that ΔAS increases the pool of immature B-cell that may be prone to lymphomagenesis.

We thank the reviewer for this useful comment, which we integrated in the Discussion of the revised manuscript. Ackr4 was shown to regulate B cell differentiation (Kara at al. (2018) J Exp Med 215, 801–813), so this is indeed one of the possible mechanisms by which a deregulation of the p53-Ackr4 axis might promote lymphomagenesis. We now mention: “Ackr4 regulates B cell differentiation (Kara et al., 2018), which raises the possibility that an altered p53-Ackr4 pathway in p53ΔAS/ΔAS Eμ-Myc male splenic cells might contribute to increase the pools of pre-B and immature B cells that may be prone to lymphomagenesis.” This is presented as one of three possible mechanisms by which decreased Ackr4 levels may promote tumorigenesis, the two others being the impact of Ackr4 on the chemokine-guided migration of lymphoma cells and its apparent effect on Myc signalling.

(3) The concordance with a male-specific prognostic effect of Ackr4 is most interesting in itself but is only of correlative evidence with respect to the study. Is there any information on whether p53AS expression is also a prognostic factor in BL? And is there evidence that Ackr4 may also be a male-specific prognostic factor in other B-cell malignancies, e.g. Multiple Myeloma?

We have now performed the CRISPR-mediated knock-out of ACKR4 in Burkitt lymphoma cells and found that it leads to a dramatic increase in chemokine-guided cell migration, which goes beyond correlation. This significant new result is mentioned in the revised abstract and presented in detail in Figures 4F and S5-S7.

Regarding p53-AS isoforms, they are murine-specific isoforms (Marcel et al. (2011) Cell Death Diff 18, 1815-1824), so there is no information on p53-AS expression in Burkitt lymphoma. Human p53 isoforms with alternative C-terminal domains are p53b and p53g isoforms, but the datasets we analyzed did not provide any information on the relative levels of p53a (the canonical isoform), p53b or p53g isoforms. We agree with the referee that this is an interesting question, but that cannot be answered with currently available datasets.

Regarding the different types of B-cell malignancies, we had already shown that Ackr4 is a male-specific prognostic factor in Burkitt lymphomas but not in Diffuse Large B cell lymphomas, which indicated that it is not a prognostic factor in all types of B cell lymphomas. For this revision, we also searched for its potential prognostic value in multiple myeloma, and found that, as for DLBCL, it is not a prognostic factor in this cancer type. This new analysis is presented in Figure S4C.